# Biomimetic Scaffolds of Calcium-Based Materials for Bone Regeneration

**DOI:** 10.3390/biomimetics9090511

**Published:** 2024-08-24

**Authors:** Ki Ha Min, Dong Hyun Kim, Koung Hee Kim, Joo-Hyung Seo, Seung Pil Pack

**Affiliations:** 1Institute of Industrial Technology, Korea University, Sejong 30019, Republic of Korea; alsrlgk@gmail.com; 2Department of Biotechnology and Bioinformatics, Korea University, Sejong 30019, Republic of Korea; jklehdgus@korea.ac.kr (D.H.K.); wood1018@korea.ac.kr (K.H.K.); sjh0413@korea.ac.kr (J.-H.S.)

**Keywords:** calcium carbonate, calcium phosphate, calcium silicate, biomedical application, bone regeneration

## Abstract

Calcium-based materials, such as calcium carbonate, calcium phosphate, and calcium silicate, have attracted significant attention in biomedical research, owing to their unique physicochemical properties and versatile applications. The distinctive characteristics of these materials, including their inherent biocompatibility and tunable structures, hold significant promise for applications in bone regeneration and tissue engineering. This review explores the biomedical applications of calcium-containing materials, particularly for bone regeneration. Their remarkable biocompatibility, tunable nanostructures, and multifaceted functionalities make them pivotal for advancing regenerative medicine, drug delivery system, and biomimetic scaffold applications. The evolving landscape of biomedical research continues to uncover new possibilities, positioning calcium-based materials as key contributors to the next generation of innovative biomaterial scaffolds.

## 1. Introduction

In the ever-evolving landscape of biomedical research, the pursuit of innovative approaches to regenerate damaged bone tissue is increasingly recognized as a priority. In this context, biomedical applications, particularly in bone tissue engineering, offer a promising avenue for addressing such challenges [1,2,3,4]. Biomaterial-based strategies have emerged as a viable alternative in various forms by providing scaffolds that support cell adhesion, proliferation, and differentiation, as well as promote the controlled release of growth factors important for tissue engineering [5,6,7]. Although the combination of calcium and phosphorus itself is not a composite, biomineral-based inorganic composites combine these mineral components to yield a chemical structure similar to that of bone tissue, promoting bone formation and growth [8,9,10,11,12,13]. These composites also play pivotal roles in tissue engineering and are widely used in clinical applications [14,15]. Among biomaterials, calcium-based materials have garnered considerable attention because of their exceptional biocompatibility and semblance to natural bone structures [16,17,18]. This attention is driven by their capacity to create a conducive environment for tissue engineering, thus making them a focal point for multidisciplinary applications in tissue engineering [4,19,20,21,22]. However, further research is needed to elucidate their potential and optimize their effectiveness in clinical settings.

Bone tissue engineering offers a promising approach for regenerating damaged bone tissue, with biomaterial-based strategies emerging as viable alternatives to traditional methods. Biomaterials act as scaffolds to support cell adhesion, proliferation, and differentiation, while facilitating the controlled release of growth factors for bone regeneration [23,24]. The key properties of biomaterials include biocompatibility, biodegradability, osteoconductivity, and osteoinductivity [25]. These materials are classified as calcium phosphate (CaP)-based, metallic, polymeric, and composite biomaterials, with calcium-based biomaterials receiving significant attention owing to their exceptional biocompatibility and similarity to natural bone [23,26].

Calcium compounds play crucial roles in tissue engineering and bone regeneration. First, calcium carbonate (CaCO_3_) enhances biocompatibility by improving interactions with osteoblasts, thereby inducing bone formation [27,28]. Second, calcium phosphate, a primary inorganic component of bones and teeth, demonstrates biocompatibility, confirming its suitability as an artificial bone material for bone regeneration [17,29,30,31]. As calcium and phosphate elements, essential minerals used as building blocks, assist in the formation of new bones, calcium phosphate has been actively researched in regenerative medicine and tissue engineering [10,30]. Third, calcium silicate, in addition to its use as an insulating and heat-preserving material, possesses biocompatibility, contributing to bone regeneration support [3,32,33]. To maintain stability within the biological environment and enhance interactions with bone tissues, the characteristics of calcium silicate in bone regeneration are being explored in the fields of regenerative medicine and tissue engineering [4,16,34]. These scholarly investigations and experimental findings underscore the importance of exploring the applicability of calcium compounds in the field of bone regeneration [26,35].

This review covers the characteristics of calcium-based materials, such as calcium carbonate, calcium phosphate, and calcium silicate, and advancements in biomaterial scaffold applications. In particular, this review focuses on research exploiting the three calcium-based materials in recent research papers. Moreover, multidisciplinary applications leveraging the properties of calcium-based materials in the fields of bone regeneration and tissue engineering have been compiled. This review addresses significant aspects of and potential future directions for exploring the use of calcium-based materials in biomimetic scaffold applications. The review is structured into three main sections: an introduction to three calcium-based materials, the latest biomedical trends related to these materials, and, finally, future perspectives and conclusions.

## 2. Calcium-Based Materials

Calcium-based materials for biomaterial application include calcium carbonates, silicates, and phosphates. These materials can exhibit nanoscale structures or nanoporosity, which are able to impart unique physical, chemical, or mechanical properties and specific functionalities. Owing to tailored biodegradability and excellent biocompatibility, calcium-based materials have been extensively studied for various biomaterial scaffold applications (Table 1).

The mechanical properties of calcium-based scaffolds frequently diverge from those of natural bone, posing significant limitations to their application in load-bearing scenarios. Specifically, calcium-based scaffolds demonstrate inferior mechanical characteristics compared to natural bone, particularly in terms of compressive strength, elastic modulus, tensile strength, fracture toughness, and fatigue resistance. Despite their advantageous biocompatibility and osteoconductivity, which make them promising candidates for bone regeneration, these mechanical shortcomings often constrain their effective use in load-bearing applications [36,37].

The synthesis methods for calcium carbonate, calcium phosphate, and calcium silicate each face specific limitations. For calcium carbonate, challenges include controlling the crystal morphology and size, managing environmental conditions, and addressing scalability and cost issues. Calcium phosphate synthesis encounters difficulties in controlling the phase and composition, managing sintering processes, and ensuring purity. Meanwhile, the synthesis of calcium silicate is limited by the need to manage reaction conditions and by-products, control the microstructure, and address the high energy and cost requirements associated with the process [10].

### 2.1. Calcium Carbonate

Calcium carbonate has attracted significant attention from both scientific researchers and industrial practitioners, owing to its versatility, design options, wide range of applications, and biocompatibility [19,38] (Figure 1).

Calcium carbonate exists in three distinct crystalline polymorphs: calcite, aragonite, and vaterite (listed in descending order of stability) [39]. These crystalline phases exhibit different crystal structures and morphologies. The crystals of vaterite, aragonite, and calcite are hexagonal, orthorhombic, and rhombohedral, respectively [40,41]. Calcium carbonate precipitation is usually divided into three main stages based on extensive studies of the precipitation mechanism: (i) the initial nucleation of amorphous calcium carbonate (ACC); (ii) the dissolution of unstable ACC, followed by recrystallization into vaterite and calcite; and (iii) the dissolution of metastable vaterite, followed by recrystallization of the most stable crystal, calcite [42]. Various synthesis variables, including temperature, pH, saturation levels, and additive concentrations, can be controlled to synthesize vaterite, aragonite, and other polymorphic forms [43,44].

The main synthesis methods for calcium carbonate particles are the precipitation method [43] and the CO_2_ bubbling method [44,45] (Table 2). The methods aim to replicate the ability of nature to control the size, shape, and phase of calcium carbonate. This involves the use of organic compounds as templates or growth modifiers, along with the consideration of various physiological parameters. The CO_2_ bubbling method has emerged as a primary approach for industrial-scale production. The CO_2_ bubbling method involves the introduction of carbon dioxide gas into an aqueous solution of calcium ions, typically in the form of calcium hydroxide (Ca(OH)_2_) or calcium chloride (CaCl_2_). In this process, carbon dioxide is bubbled through the calcium-containing solution under controlled conditions. The concentration of CO_2_, the pH of the solution, and the reaction temperature are crucial parameters that influence the nucleation and growth of calcium carbonate crystals [46,47,48].

Solution precipitation is the most established method for preparing calcium carbonate nanoparticles (NPs) and involves the reaction between Ca^2+^ and CO_3_^2−^ in an aqueous solution. This method allows for the production of large quantities of calcium carbonate NPs without the need for a surfactant, thereby reducing production costs. The mild preparation conditions provide the advantage of allowing various bioactive species, such as small-molecule drugs, genes, and proteins, to be loaded into calcium carbonate NPs during the precipitation process [49]. Significantly, synthesis variables, such as pH, temperature, ion concentration, stirring speed, solvent species, and additives, are often manipulated to control the size, shape, and phase of calcium carbonate NPs [50]. The gas diffusion method is primarily used to prepare ACC loaded with small-molecule drugs [45]. In the solution precipitation method, CaCl_2_ is dissolved in ethanol and transferred to a glass bottle. Subsequently, the bottle is placed in a desiccator, alongside another bottle that contains ammonium bicarbonate. CO_2_ and NH_3_ are generated from ammonium bicarbonate and then dissolved in an ethanol solution to form CO_3_^2−^ and NH^4+^. Under alkaline conditions induced by NH^4+^, CO_3_^2−^ reacts with Ca^2+^ to form ACC. This method offers control over the size, shape, and polymorph of the prepared ACC by adjusting the additives, temperature, and Ca^2+^ concentration [51].

### 2.2. Calcium Phosphate 

Similar to calcium carbonate nanomaterials, calcium-phosphate-based nanomaterials have several advantages, including controlled delivery of drugs, improvement of tissue integration, ease of fabrication, and scalability [10] (Figure 2). Calcium phosphate scaffolds, with their high surface-area-to-volume ratio, exhibit unique properties beneficial for bone tissue engineering, facilitating enhanced cell adhesion and proliferation [52]. Additionally, they promote cell differentiation and enable the loading of therapeutic ingredients [53]. Furthermore, calcium phosphate can improve mechanical properties and facilitate controlled drug release [54,55,56]. Consequently, calcium phosphate nanomaterials have garnered increasing attention, leading to the development of various preparation strategies to meet clinical requirements.

Various methods have been devised to produce calcium phosphate NPs of various sizes, shapes, and compositions [57]. These techniques are specifically designed to create calcium phosphate NPs, typically smaller than 100 nm, that exhibit varying degrees of agglomeration for use in biological and medical applications [58]. The chemical composition of calcium phosphates includes multiple ions, including calcium (Ca^2+^), orthophosphate (PO_4_^3−^), metaphosphate (PO_3_^−^), pyrophosphate (P_2_O_7_^4−^), and hydroxide (OH^−^) [59] (Table 3). The size and crystallinity of the precipitated particles can be finely adjusted by manipulating variables such as pH, concentration, temperature, and precipitation duration, offering a significant advantage in tailoring the material properties for specific applications. Rapid precipitation at room temperature often yields particles with a low crystalline structure, whereas improved particle crystallinity and shape definition can be achieved by both increasing the temperature and slowing the rate of crystallization [60,61]. Moreover, in contrast to high-temperature methods of synthesis involving organic solvents, which may denature biomolecules, such as nucleic acids and proteins, the addition of bio-organic chemicals during the synthesis process ensures compatibility with biological systems, while retaining the desired material properties [62]. This capability enhances the potential biomedical applications of calcium phosphate NPs.

Various techniques have been used for the fabrication of calcium phosphate NPs, including wet-chemical precipitation [63], sonochemical synthesis [64], enzyme-assisted methods [65], solvothermal synthesis [66], sol–gel methods [67], microwave-assisted methods [68], spray-drying [69], and electrospinning [70]. Among these methods, precipitation is the most frequently used for the production of nanosized calcium-phosphate-based biomaterials [71]. Additionally, calcium-phosphate-based nanomaterials offer versatility in their customization by allowing the adjustment of the parameters of variables, such as size, shape, and surface chemistry [72]. Functionalized calcium phosphate NPs exhibit various properties, such as osteogenic, antimicrobial, and bioimaging capabilities [10].

The precipitation method is a straightforward approach to the synthesis of hydroxyapatite (HA). By adjusting the parameters of processing variables, such as pH, solvent type, ion concentration, reaction time, and additives, control over the particle size, phase, and structure can be achieved with ease [61]. This technique is the predominant strategy for producing calcium phosphate NPs, owing to its convenience, adaptability, and simplicity. HA exhibits minimal solubility in aqueous solutions at ambient temperature and has a pH of approximately 4.2, making it typically the most stable phase of calcium phosphate [20,73].

### 2.3. Calcium Silicate

Calcium silicate has diverse applications in the field of bone regeneration because of its high safety and biocompatibility [4,16,74]. Moreover, calcium silicate is a suitable material for bone grafts and supplements (Figure 3). Its biocompatibility and ability to promote interactions with bone tissues facilitate bone formation, making it a valuable component for addressing bone damage or deficiencies through grafting procedures [75].

Commonly used calcium silicate materials include calcium metasilicate (CaSiO_3_), ortho-calcium silicate (Ca_2_SiO_4_), and tricalcium silicate (Ca_3_SiO_5_). Recent investigations have underscored the remarkable bioactivity and biocompatibility of calcium silicate ceramics, such as dicalcium silicate (Ca_2_SiO_4_) [76,77], pseudowollastonite (β-CaSiO_3_) [78], and wollastonite (α-calcium silicateO_3_) [79] (Table 4). The composition, amount, crystallinity, and morphology of the synthesis product are influenced by the process conditions (temperature, speed of reagents mixing, etc.) and the medium composition (component concentration, pH). Consequently, several ceramics based on calcium silicate are considered promising candidates for artificial bone substitutes. Furthermore, calcium-silicate-based materials can release calcium (Ca^2+^) and silicon (Si^4+^) ions within the physiological milieu. Notably, Ca^2+^ ions play a pivotal role in promoting the proliferation and differentiation of osteoblasts and osteoclasts [80]. They modulate osteogenesis by stimulating angiogenesis, facilitating the release of growth factors, augmenting bone density, averting osteoporosis, and bolstering cell proliferation and bone mineralization [81,82]. Conversely, Si^4+^ ions serve as bioactive constituents within the human body, orchestrating osteogenic differentiation of bone marrow mesenchymal stem cells and engaging in the mineralization process of nascent bone formation. These observations underscore the potential of calcium silicate in enhancing osteogenesis and facilitating bone regeneration [83,84,85].

Owing to the persistent efforts of researchers, an expanding array of synthetic methods is being developed and refined via methods like high-temperature synthesis involving solid-phase reactions [87], wet synthesis using co-precipitation [88], the sol–gel method [89], the hydrothermal method [90], and solution combustion methods [91]. The sol–gel method, initially used for synthesizing calcium silicate, offers advantages such as straightforward preparation, a controlled composition, high homogeneity of mixing, a low reaction temperature, and high purity [92]. In recent years, researchers have made significant advancements in raw material sourcing and synthesis methods, leading to the emergence of several innovative green synthesis approaches [93].

In the dental field, calcium silicate is used as a material for light-curing treatments and root canal synthesis. This application aids in the protection and regeneration of bone-related dental tissues during dental procedures [94]. Calcium silicate is considered biologically safe and is used in various biomimetic scaffold materials and implants. This usage contributes to the formation of artificial bones and related tissues, promoting the regeneration of damaged or missing bone structures [95]. Furthermore, calcium silicate plays an important role in tissue engineering and cell culture applications. Its use enhances interactions with cells and facilitates cell proliferation and differentiation, making it valuable in tissue regeneration and cell culture [16]. The versatility of calcium silicate and its positive impact on bone regeneration make it a valuable material, with applications ranging from insulation to dental procedures and tissue engineering, highlighting its potential across diverse biomimetic scaffold and tissue engineering applications.

## 3. Calcium-Based Materials for Biomedical Applications

Calcium-based materials, such as calcium carbonate, calcium phosphate, and calcium silicate, are used in various industries, particularly in biomedical applications, materials engineering, and chemical fields. Calcium-based scaffolds support bone formation and remodeling in different ways each. Calcium carbonate enhances osteoblast activity through calcium ion release and is resorbed by osteoclasts during bone remodeling. Calcium phosphate, including hydroxyapatite and tricalcium phosphate, supports osteoblast functions and is resorbed by osteoclasts, promoting balanced bone remodeling. Calcium silicate promotes osteoblast activity through the release of calcium and silicon ions and is gradually replaced by natural bone as osteoclasts resorb it [96,97].

Current research has focused on using the unique properties of these materials for a variety of applications, including drug delivery, biomimetic scaffold materials, bone regeneration, and tissue engineering.

### 3.1. Calcium-Carbonate-Based Applications

Calcium carbonate has been used as a bone substitute scaffold or composite, owing to its osteoconductive properties. Calcium carbonate is used in bone regeneration for its biocompatibility and ability to provide a scaffold for new bone growth, though its effectiveness varies with its crystalline form. Calcite, the most stable form, features a rhombohedral structure and offers high mechanical strength, making it suitable for scaffolding, although its low solubility may impact integration and resorption over time. Aragonite, with an orthorhombic structure, is less stable than calcite but more soluble, which can facilitate better integration with surrounding bone and promote new bone formation, due to its greater resorbability. Vaterite, the least stable and most soluble form, has a hexagonal structure and is typically used in experimental settings; its rapid dissolution under physiological conditions allows for quick resorption and potential for faster bone regeneration, but its instability may limit any long-term structural support it provides [86,98,99,100,101]. Table 5 outlines some of the biomedical applications of calcium carbonate in various fields, particularly bone research. Calcium carbonate coatings can help to impart biological properties to non-biological materials. These applications provide a conducive environment for bone regeneration. Furthermore, calcium carbonate can serve as a drug delivery carrier because it responds to acidic pH, causing dissolution [38,42]. Antibiotics can be loaded to address osteomyelitis, whereas therapeutic agents can be incorporated to enhance overall bone mass. In the field of scaffolds, calcium carbonate is mainly used in the form of vaterite. Chitosan and polyethylene glycol can be used in combination, as long as they do not cause toxicity. These scaffolds are used to repair various bone parts, such as the calvaria, the skull, and the forearm. Vaterite’s application areas in drug delivery systems can be as diverse as osteoporosis, bone substitutes, osteogenesis, antibacterial effects, and osteomyelitis. Vaterite is also used because its porosity is important for drug retention. Various strategies are used to improve its drug-loading capacity in various in vitro and in vivo models. Although there are relatively few applications of calcium carbonate in coatings, it is mainly used in implant applications.

The majority of hydrogel scaffolds used for bone regeneration fail to meet clinical requirements because of their uncontrolled mechanical properties, insufficient calcium delivery, and lack of significant osteogenesis (Figure 4A). To address these shortcomings, a novel composite acid-responsive hydrogel scaffold platform was developed to provide controlled stiffness and calcium delivery to enhance bone regeneration [105]. F127-DA (Pluronic F127 diacrylate) was chosen as the scaffold matrix material, owing to its excellent biocompatibility and ease of formulation. The composite structure with a low amount of nano-CaCO_3_ exhibited excellent mechanical, swelling, and degradation properties similar to the composite structure. In F-Ca-A hydrogels, nano-CaCO_3_ was situated in the core region and could be gradually released at the bone defect site, facilitating the expression of osteogenesis-related genes and mineralization. Moreover, the hydrogel scaffold efficiently modulated the expression of key markers, such as bone morphogenetic protein 2 (BMP2), collagen type 1 alpha 1 (Col1), and osteopontin (OPN), which are crucial for promoting osteogenesis. Moreover, in vivo experiments revealed that the F-Ca-A hydrogel resulted in superior bone regeneration compared to the F-Ca hydrogel and F hydrogel. This result is potentially attributed to the gradient distribution of calcium ions within the hydrogel matrix induced by the acid response. This gradient distribution spatially induces osteoblast differentiation and migration into the hydrogel matrix, thus facilitating the formation of new bone.

In the application of drug delivery systems, a recent publication describes a microenvironment-responsive nanoplatform denoted as HMCZP, comprising minocycline-modified hyaluronic acid/calcium carbonate/zoledronic acid@succinic anhydride (SA)-modified poly(β-amino ester) [109] (Figure 4B). HMCZP shows significant promise in inhibiting osteoclast viability, differentiation, and function, providing a potential treatment for catabolically active osteoporosis by using the therapeutic potential of zoledronic acid and focusing on the extracellular acidic microenvironment. The results of pH-responsive and H+ consumption features of HMCZP show that it could potently inhibit both osteoclast differentiation and resorption activity. The inhibition of osteoclast function by HMCZP is linked to osteoclast activity. HMCZP demonstrates an effect on inhibiting bone mass loss in regions exhibiting high-bone-remodeling activity. In terms of osteogenesis, HMCZP can suppress excessive bone resorption without affecting osteogenesis. Moreover, HMCZP suppresses the expression of genes associated with osteoclast differentiation (Trem2, TRAP, and Fcgr3a), suggesting its potential utility as a biomarker or a therapeutic target for the treatment of osteoporosis.

A recent study detailed the development of innovative calcium carbonate coatings using the hydrothermal method, which was subsequently enhanced by the addition of either glutamic acid or dopamine to improve the degradation and biological performance of the MgZnCa alloy [115] (Figure 4C). This study explored the potential of calcium carbonate coatings for improving the performance of magnesium alloy implants for biomimetic scaffold applications. According to the results, organic molecules, particularly Glu, stabilize vaterite and hinder rod-like aragonite deposition, resulting in more uniform and thicker coatings with stronger adhesion. Vaterite-dominated coatings enhance the in vitro corrosion resistance and degradation performance of Mg alloys. In vitro cell culture tests confirm good cytocompatibility for CaCO_3_-based coatings. Moreover, the improved in vitro osteogenic performance of osteoblasts and accelerated in vivo bone regeneration with CaCO_3_-Glu hybrid coatings suggest enhanced biological activity, particularly with vaterite. Furthermore, the study demonstrated favorable cytocompatibility and promising in vitro osteogenic activity among osteoblasts, indicating their potential for enhancing the clinical applicability of magnesium alloy implants.

### 3.2. Calcium-Phosphate-Based Applications

Calcium phosphate is an environmentally friendly, biodegradable material that shares chemical similarities with human hard tissues, such as bone and teeth, making it highly biocompatible (Table 6). Calcium phosphates exhibit exceptional biological properties, are cost-effective and straightforward to manufacture, and are generally considered safe. Consequently, they may receive clinical approval relatively quickly for various applications [9,20,24]. Calcium phosphate has similar applications to calcium carbonate. In terms of scaffold applications, it is being used for bone regeneration and as a bone substitute and is being used for overall bone tissue engineering. Different types of calcium phosphates are used in combination with both organic and inorganic materials. Drug delivery systems and coatings are also being applied to bone tissue engineering in the form of various organic and inorganic complexes.

The literature details the incorporation of HA NPs, synthesized directly on carbon nanotubes (CNTs), to strengthen poly(ε-caprolactone) (PCL) bone scaffolds [122] (Figure 5A). This method seeks to merge the superior mechanical attributes of CNTs with the biological activity of HA to advance bone regeneration efforts. Research has delved into the nucleation and expansion of HA NPs on CNTs, assessing how these composites might bolster the physical and mechanical traits of PCL bone scaffolds. Moreover, efforts have concentrated on the in situ synthesis of HA NPs on carboxylated multi-walled CNTs to avert their clumping within PCL scaffolds produced through selective laser sintering (SLS). The mechanism of in situ synthesis of HA nanoparticles involves carboxyl functional groups of cCNT serving as anchor sites for Ca^2+^ deposition, followed by electrovalent bonding with HPO_4_^2−^ for HA nucleation. The resulting HA nanoparticles, ranging from 10 to 30 nm in diameter and 40–80 nm in length, are uniformly generated on the cCNT surface. Incorporating 12% cCNT-HA into PCL scaffolds significantly increases tensile and compressive strengths by 86.6% and 31.9%, respectively, attributed to the bridging and pulling out of cCNT-HA. Mineralization experiments show that the scaffold’s apatite layer has a Ca/P ratio close to that of human bone tissue, indicating good bioactivity for apatite mineralization induction. Moreover, the scaffolds exhibit favorable cytocompatibility, supporting cell attachment, growth, and proliferation.

A recent study introduced triple-functionalized calcium phosphate NPs as an advanced drug delivery system for the regeneration of bone tissue [131] (Figure 5B). According to the experimental findings, the application of triple-functionalized NPs significantly accelerates bone healing. The study used gene manipulation techniques with CaP nanoparticles to enhance bone healing in rat femoral head defects. DNA-loaded nanoparticles promoted BMP-7 and vascular endothelial growth factor synthesis, while siRNA-loaded ones inhibited TNF-a. Both DNA- and siRNA-loaded pastes significantly improved bone healing compared to controls. A combined paste containing DNA for BMP-7 and vascular endothelial growth factor, along with siRNA against TNF-α, showed the most accelerated bone healing. The use of calcium phosphate NPs for drug delivery resulted in significant increases in bone formation at both protein and gene levels (OCN, Runx2, and SP7) after 10 days. Additionally, it significantly accelerated new bone formation after 21 days compared to other treatments. These results demonstrated the potential effectiveness of triple-functionalized calcium phosphate NPs in bone tissue repair.

Although polyether ether ketone (PEEK) has gained widespread interest in orthopedics, its inherent biological inertness and limited osteogenic properties have limited its utility in the treatment of bone tumors (Figure 5C). To overcome this challenge, Shu et al. developed novel PEEK scaffolds modified with HA NPs and molybdenum disulfide (MoS_2_) nanosheets using a hydrothermal technique [130]. The PEEK scaffolds have innovative dual-effect synergistic properties that give them exceptional photothermal therapeutic capabilities. These capabilities are dependent on the concentration of molybdenum ions (Mo^2+^) and laser power density, and they exceed those of conventional PEEK scaffolds. MoS_2_ autonomously nucleates and diffuses from PEEK scaffolds, prompting the growth of flower-like HA crystals on its surface. This incorporation of MoS_2_ enhances the scaffolds’ photothermal properties, enabling effective tumor treatment through photothermal therapy (PTT). Additionally, HA modification improves the scaffolds’ osteogenic induction capability and serves as a source of calcium and phosphorus for matrix mineralization. In vivo analysis of rat femora treated for 4 weeks using micro-computed tomography and histological methods further confirmed the exceptional photothermal and osteogenic capabilities of the 3D-printed modified scaffolds.

In nanostructure calcium phosphate for bone regeneration, nanohydroxyapatite, due to its similarity to natural bone mineral, has been widely researched. Its high surface area and porosity enhance cell attachment, proliferation, and differentiation, making it an excellent scaffold material for bone regeneration [10]. Recent studies have focused on HA composites with polymers and other bioceramics to improve mechanical strength and biological performance [132]. Innovations in biodegradable scaffolds have incorporated calcium phosphate to provide temporary support for new bone growth. These scaffolds gradually degrade as new bone tissue forms, eliminating the need for surgical removal.

### 3.3. Calcium-Silicate-Based Applications

Calcium silicate is a bioceramic commonly used to repair hard tissues, such as bone and teeth. In the field of drug delivery, calcium silicate is often used to load antibiotics to inhibit infections caused by external substances or wounds. This feature allows calcium silicate to function as a scaffold on its own or as a coating on structures, such as titanium, to enhance bioactivity [4,33,34]. Table 7 summarizes studies that have used calcium silicate as a biomedical material for dental and orthopedic applications. Similar to calcium carbonate and calcium phosphate, calcium silicate has been extensively investigated in various in vivo models, demonstrating its efficacy. It can be formulated with various inorganic materials to increase cell adhesion or enhance regeneration. Scaffolds can be created and applied to bone cement or implants, but they can also be used with drug delivery systems.

The study introduced a novel approach using OPN sequence-modified mesoporous calcium silicate (MCS) scaffolds to enhance angiogenesis during bone tissue regeneration [134] (Figure 6A). Through detailed experimentation, the researchers found that the modification of MCS scaffolds with the OPN sequence led to a notable increase in the expression of vascular endothelial growth factor (VEGF), a key protein involved in angiogenesis. Moreover, in vitro experiments demonstrated that endothelial cells exhibit higher rates of proliferation and migration when cultured on OPN-modified scaffolds than on unmodified scaffolds, indicating enhanced angiogenic potential. Furthermore, in vivo experiments conducted on animal models revealed a notable increase in blood vessel formation and vascularization within the bone defect area when OPN-modified scaffolds were used. This enhanced vascularization contributes to improved bone tissue regeneration, as evidenced by quantitative measurements of bone volume and histological assessments showing more mature and organized bone tissue formation. Overall, these findings highlight the potential of OPN-modified MCS scaffolds as a promising strategy for promoting angiogenesis and enhancing bone tissue regeneration.

Notably, investigators applied a bioactive coating of magnesium calcium silicate to the surface of PEEK, and two types of polyphenols, genistein and curcumin, were loaded together into the PEEK coating [144] (Figure 6B). The researchers explored the application of bioactive coatings on PEEK surfaces for dual-drug release, with the aim of enhancing antibacterial activity, responses of rat bone marrow mesenchymal stem cells (rBMSCs), and osteointegration. The experimental results demonstrated that the dual-drug-release system exhibited superior antibacterial activity compared to metallic coatings and single-drug-release systems. Additionally, it promoted the differentiation and secretion of rBMSCs, thereby positively impacting bone regeneration. Furthermore, the bioactive coating enhanced the surface compatibility of PEEK materials and improved vascular neogenesis for enhanced implant-tissue integration and osseointegration.

In coating applications, europium (Eu) has garnered significant interest as a means of introducing biologically active ions, thereby imparting diverse biological and functional characteristics to biomaterials [146] (Figure 6C). In one study, calcium silicate coatings incorporating varying concentrations of Eu were successfully applied onto titanium substrates using electrophoretic deposition. Notably, a lower Eu content (2.5 mol%) yielded coatings with higher density and enhanced adhesion strength (~3.3 N). All Eu–calcium silicate coatings exhibited a favorable apatite-forming ability, albeit with a reduced degradation rate compared to the calcium silicate coatings alone. Bioactivity of all Eu–CS coatings was sufficient to form apatite, but their degradation rate was slightly lower than those of CS coatings. The doping of Eu into CS caused a spherulite structure to form, while CS formed uneven layers. After immersion in SBF for 7 days, the Eu2.5CS coating induced apatite with a Ca/P ratio that is close to that of biological apatite. Of particular interest, Eu2.5calcium silicate demonstrated cell proliferation comparable to calcium silicate coatings and augmented the osteogenic activity of the calcium silicate coating, suggesting its potential as a promising biomaterial for bone tissue engineering applications.

## 4. Challenges and Future Perspectives

Recent studies of bone regeneration and tissue engineering applications have shown that significant progress has been made in the development and application of calcium-based materials in bone regeneration and tissue engineering. These advancements primarily focus on improving material properties, biocompatibility, and functional performance [23,26]. Considering the recent research on nanostructured calcium-based scaffolds, electrospun (electrospinning) nanofibers incorporating calcium-based nanoparticles have shown promise in mimicking the extracellular matrix of bone. These nanofibers support the growth and differentiation of osteogenic cells, facilitating bone regeneration. The use of electrospinning allows for the creation of highly porous and interconnected structures that enhance nutrient flow and cell migration [70]. The use of 3D printing technology to create custom scaffolds with calcium-based materials has seen significant advances. This approach allows for the precise fabrication of scaffolds tailored to patient-specific needs, enhancing the integration and functionality of the implants [147].

Each calcium-based materials presents specific biocompatibility issues in physiological environments. Calcium carbonate can degrade quickly, especially under acidic conditions, potentially leading to an inflammatory response from its degradation products. Calcium phosphate, which includes phases like hydroxyapatite (HA) and β-tricalcium phosphate (β-TCP), varies in stability and degradation rates, with high-porosity forms potentially being mechanically weaker. Calcium silicate materials can also degrade under physiological conditions, releasing silica and calcium ions, which poses challenges for maintaining long-term stability, as their properties may change over time [4,6,17]. Addressing these biocompatibility concerns through material modifications, improved synthesis methods, and a better understanding of material–biological interactions is crucial for enhancing their safety and effectiveness in long-term biomedical applications.

The future of calcium-based materials in biomedical applications holds numerous exciting possibilities. Firstly, these materials hold promise for the development of advanced drug delivery systems featuring controlled-release kinetics. This suggests that smart and targeted delivery systems can be designed to achieve more precise and effective therapeutic outcomes [95,126,131]. Secondly, the ability to tailor nanostructures of calcium-based materials for specific biomedical applications is a noteworthy prospect. The customization of these materials based on their intended use, whether for bone regeneration, drug delivery, or other purposes, offers a versatile approach to addressing a variety of biomedical challenges [18,80,106]. Another avenue of exploration involves the use of combination therapies. Future applications may involve the integration of calcium-based materials with other biomaterials or therapeutic agents. This synergistic approach has the potential to significantly enhance overall treatment efficacy, opening up new possibilities for multifaceted biomedical interventions [70,84,113,145]. Moreover, a crucial future perspective is the translation of these materials from laboratory research to clinical application. Rigorous preclinical studies and safety concerns are paramount for the successful clinical integration of calcium-based materials. Exciting advancements in 3D printing technologies offer a platform for the precise design and fabrication of complex structures using calcium-based materials. These innovations open up novel possibilities in tissue engineering and regenerative medicine, allowing the creation of intricate structures tailored to specific biomedical needs [1,77,136].

Nanostructures significantly increase the surface area compared to non-nanoporous materials. This provides more space for cell attachment, proliferation, and differentiation, which is essential for effective bone regeneration and integration with surrounding tissues [38,80,128]. Nanopores better mimic the natural extracellular matrix (ECM) of bone tissue, promoting enhanced cell adhesion and growth. This leads to more effective tissue engineering and faster healing processes, making nanostructures superior to materials with larger pores or non-porous structures [148]. Calcium-based materials, with their high porosity, enhance cell infiltration and tissue integration by facilitating efficient nutrient delivery and waste removal, which are critical for sustaining cell viability and promoting tissue growth. Furthermore, calcium-based materials support direct bone ingrowth and integration with host bone, making them highly effective for bone regeneration; their pore size and interconnectivity are crucial for maximizing osteoconductivity [5,31,53,108]. Additionally, their porosity supports vascularization, which is essential for supplying nutrients and removing waste products from regenerating tissue. The smooth interaction at the cellular level minimizes inflammation and other adverse reactions, promoting better healing and biocompatibility [149,150].

The future perspectives for calcium-based materials have potential applications in advanced drug delivery and innovative applications using 3D printing technologies. As research progresses, these materials are poised to play pivotal roles in shaping the next generation of biomedical technologies, advancing therapeutic interventions, patient care, and overall progress in biomedical science.

## 5. Conclusions

This review explored calcium-based materials, including calcium carbonate, calcium phosphate, and calcium silicate, in view of the development of effective regeneration materials. These three types of calcium-based materials have excellent properties, such as biodegradability, bioavailability, applicable diversity, and surface modification, which are essential for biomimetic scaffold applications. Recent technological advancements have notably expanded the surgical applications of calcium carbonate, calcium phosphate, and calcium silicate biomaterials in hard-tissue surgery.

For calcium carbonate, enhancements in vaterite-based scaffolds and drug delivery systems have improved their efficacy in bone regeneration and targeted antibiotic delivery, addressing diverse clinical needs. Advanced fabrication techniques, like hydrothermal synthesis and composite formulations, have improved the stability and functionality of biomaterials. Key advancements include the development of acid-responsive hydrogels with calcium carbonate for controlled drug delivery and enhanced bone regeneration, as well as the use of functional additives, such as glutamic acid, to boost the stability and performance of calcium carbonate coatings for better bone repair and regeneration.

Calcium phosphate materials have also seen expanded use, with innovations such as the incorporation of carbon nanotubes and biodegradable polymers enhancing their performance in scaffold applications and drug delivery systems. This has broadened their roles in bone regeneration and implant coatings. Synthesis of hydroxyapatite nanoparticles on carbon nanotubes has enhanced scaffold strength and biological performance. Additionally, triple-functionalized calcium phosphate nanoparticles have notably accelerated bone healing and improved targeted drug delivery.

Similarly, advancements in calcium silicate materials, including the development of bioactive coatings and functionalized nanoparticles, such as europium-doped calcium silicate, have improved osteogenic induction and angiogenesis. These advancements have also expanded the use of calcium silicate materials in both dental and orthopedic contexts, enhancing their effectiveness in promoting bone tissue regeneration and improving implant integration. These developments highlight a significant expansion in the surgical applicability of these biomaterials, offering more effective and versatile solutions in complex hard-tissue surgeries.

## Figures and Tables

**Figure 1 biomimetics-09-00511-f001:**
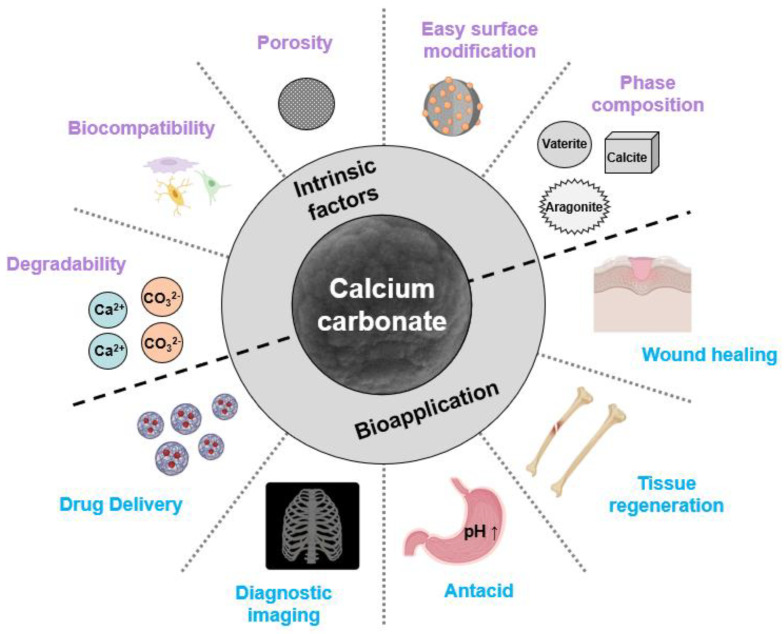
Properties of calcium carbonate nanoparticles and their biomedical applications.

**Figure 2 biomimetics-09-00511-f002:**
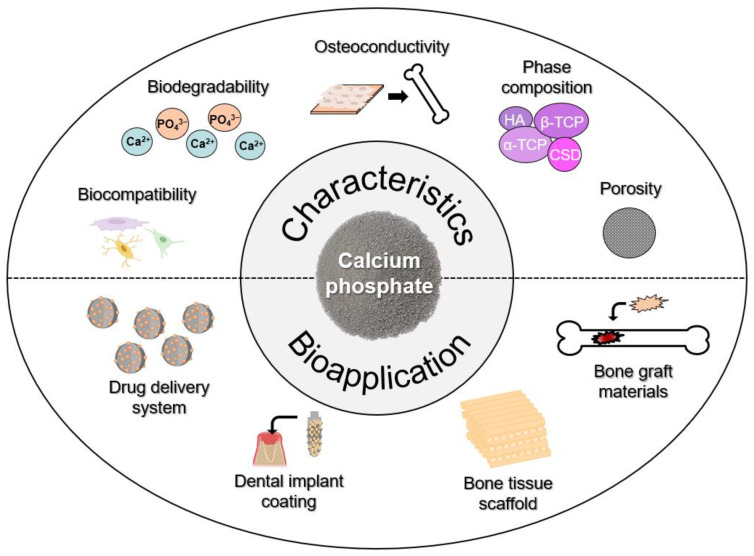
The biomaterial scaffold applications of calcium phosphate.

**Figure 3 biomimetics-09-00511-f003:**
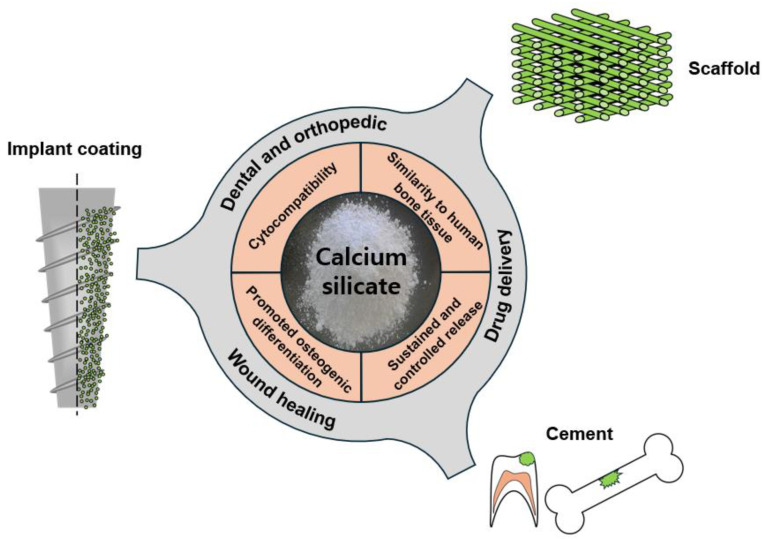
Biomaterial scaffold applications of calcium silicate focused on hard-tissue repair and engineering.

**Figure 4 biomimetics-09-00511-f004:**
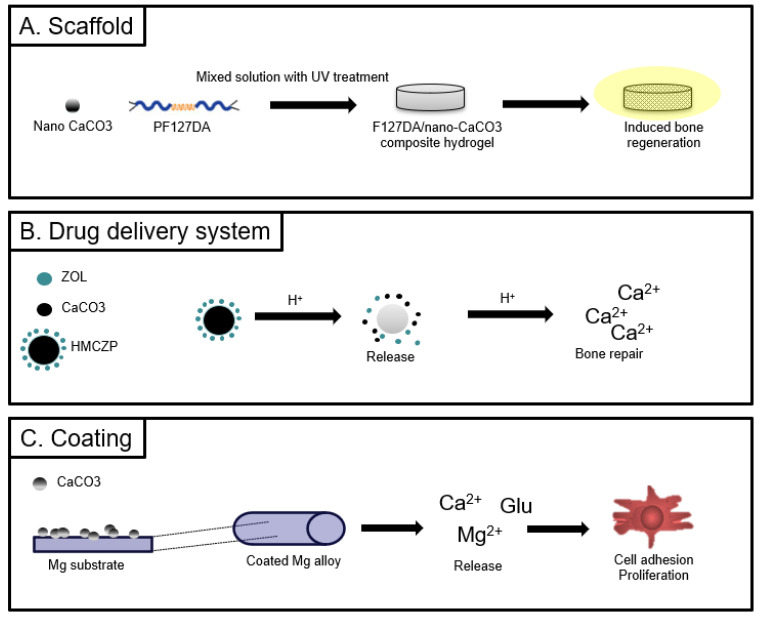
Biomedical applications of calcium carbonate focused on scaffolds [105], drug delivery [109], and coatings [115]. Schematic representation illustration of each study.

**Figure 5 biomimetics-09-00511-f005:**
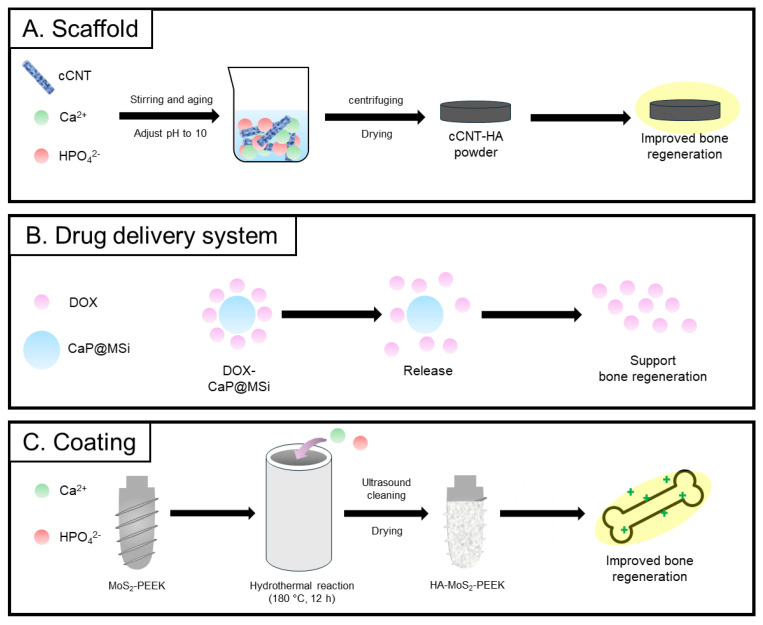
Biomedical applications of calcium phosphate focused on scaffolds [122], drug delivery [131], and coatings [132]. Schematic representation illustration of each study.

**Figure 6 biomimetics-09-00511-f006:**
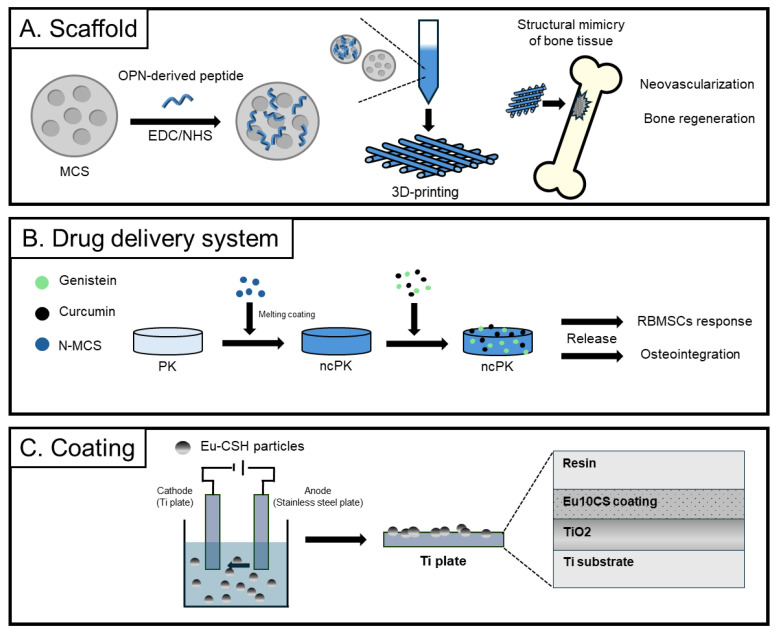
Biomedical applications of calcium silicate focused on scaffolds [134], drug delivery [144], and coatings [146]. Schematic representation illustration of each study.

**Table 1 biomimetics-09-00511-t001:** A comparative overview of the key properties of calcium carbonate, calcium phosphate, and calcium silicate.

Property	Calcium Carbonate	Calcium Phosphate	Calcium Silicate
Chemical composition	CaCO_3_	Ca_3_(PO_4_)_2_	Ca_2_SiO_4_
Biocompatibility	Generally high	Generally high	Generally high
Biodegradability	Biodegradable	Biodegradable	Biodegradable
Solubility in water	Limited solubility	Limited solubility	Insoluble
Bone mimicking	Limited bone mimicking	Exceptional	Superior
Osteoinductivity	Low	High	Moderate to high
Thermal stability	Decomposes at high temperatures	Stable at high temperatures	Stable at high temperatures
Applications	Delivery vehicles, supplements	Bone grafts, dental implants	Supplements, biomedical devices
Uses in tissue engineering	Limited applications	Mainly bone regeneration	Limited applications

**Table 2 biomimetics-09-00511-t002:** The synthesis methods for carbonate precipitation, calcium carbonate particles, and nanoparticles.

**Method**	**Pros**	**Cons**	**Challenges**
Spontaneous precipitation	Biocompatibility Versatility Ease of implementation	Limited scalability Uniformity issues Limited control over properties	Need for an additive to control the size and CaCO_3_ phase Difficulty synthesizing at an upscale level
Slow carbonation	Biocompatibility Ease of scale-up Environmental sustainability	Long processing time Complexity Regulatory considerations	Difficulty synthesizing uniform CaCO_3_ Extended synthesis time
Reverse emulsion	Controlled particle size and morphology Uniformity and monodispersed nature Encapsulation of active ingredients	Complexity of emulsion formation Limited scalability Potential for residual surfactants	Various factors controlling the size and morphology Surfactant removal steps Difficulty synthesizing at an upscale level
Hydrothermal and solvothermal synthesis	Enhanced reactivity Versatility High purity	Equipment complexity Limited solvent compatibility Safety concerns Energy-intensive nature	Optimization of reaction conditions Need for contaminant control
CO_2_ bubbling	Environmentally friendly Ease of implementation Biocompatibility	Long processing time Limited control over particle properties Scale-up challenges	Difficulty synthesizing uniform CaCO_3_ Need for contaminant control

**Table 3 biomimetics-09-00511-t003:** A comparative overview of calcium phosphate materials.

Name	Abbreviation	Chemical Formula	Ca/P Ratio
Hydroxyapatite	HA	Ca_10_(PO_4_)_6_(OH)_2_	1.67
Calcium-deficient hydroxyapatite	CDHA	Ca_10−x_(HPO_4_)_x_(PO_4_)_6−x_ (OH)_2_ (0 < x < 1)	1.5–1.67
Dicalcium phosphate dihydrate	DCPD	CaHPO_4_∙2H_2_O	1
α-Tricalcium phosphate	α-TCP	α-Ca_3_(PO_4_)_2_	1.5
β-Tricalcium phosphate	β-TCP	β-Ca_3_(PO_4_)_2_	1.5
Octacalcium phosphate	OCP	Ca_8_(HPO_4_)_2_(PO_4_)_4_·5H_2_O	1.33
Fluorapatite	FA	Ca_10_(PO_4_)F_2_	1.67

**Table 4 biomimetics-09-00511-t004:** Common sources of calcium silicate materials.

Name	Chemical Formula	Components	Chemical Equation
Calcium metasilicate	CaSiO_3_	CaO·SiO_2_	r CaX_2_ + m M_2_SiO_3_ + n H_2_O → rCaO·mSiO_2_·nH_2_O + MX [86]
Dicalcium silicate	Ca_2_SiO_4_	2CaO·SiO_2_	
Tricalcium silicate	Ca_3_SiO_5_	3CaO·SiO_2_	
Rankinite	Ca_3_Si_2_O_7_	3CaO·SiO_2_	
Tobermorite	Ca_5_Si_6_O_16_(OH)_2_·4H_2_O	5CaO·6SiO_2_·5H_2_O	
Pseudowollastonite	β-CaSiO_3_	CaO·SiO_2_	
Wollastonite	α-CaSiO_3_	CaO·SiO_2_	

**Table 5 biomimetics-09-00511-t005:** Calcium-carbonate-based biomedical research for bone regeneration applications.

Biomedical Application	Materials	Primary Function	Ca-Based Effect	Biomedical Results	Ref.
Scaffold	CaCO_3_ (calcite), chitosan	Osteogenesis	To promote a bone-like environment Calcium ions for bone regeneration	MSC migration and osteogenic differentiation ↑ Calvaria defect repair (rats)	[102]
	CaCO_3_ (eggshell derived), MgO, chitosan, BMP-2	Bone substitute	Sustainable release of MgO Calcium ions for bone regeneration	Inducing osteoinductive effects (hADSCs) Calvaria defect repair (SD rats)	[103]
	CaCO_3_ (vaterite), PEG	Bone substitute	Microsized precursor of hydroxyapatite for mineralization	Bone tissue regeneration ↑(BALB/c mice)	[104]
	CaCO_3_ (vaterite), Pluronic F127 diacrylate	Bone substitute	Acid-responsive property for spaced controlled distribution of scaffolds Calcium ions for expression of osteogenesis-related genes	Biocompatibility Osteogenic differentiation ↑ Skull defect repair (New Zealand white rabbits)	[105]
	CaCO_3_ (vaterite), Ag nanoparticle, advanced platelet-rich fibrin	Bone substitute	Calcium ions for osteoconductivity Nanosized precursor of β-tricalcium phosphate and hydroxyapatite	Forearm defect repair (New Zealand white rabbits)	[106]
Drug delivery system	CaCO_3_ (vaterite), BMP-2, GelMA	Bone substitute	pH-sensitive release of BMP-2 in the weakly acidic environment of bone injury Bone filling materials by promoting calcium ions	Skull defect repair (SD rats)	[107]
	CaCO_3_ (vaterite), β-estradiol, AC4ManNAz	Osteoporosis	Control of β-estradiol release in the acidic osteoporotic microenvironment Regulating the activity of osteoclasts and osteoblasts by promoting calcium ions Incresaing β-estradiol in vivo stability and availability by nanostructure	Osteoclast proliferation, bone resorption activity ↓ Osteoblast proliferation, differentiation, bone generation ↑ Osteoporosis treatment (C57BL/6J mice)	[108]
	CaCO_3_ (vaterite), HA-MC, ZOL, PBAE-SA	Osteoporosis	Synergistically enhanced antiosteoporotic effects on zoledronic acid H^+^ consumption for inhibiting osteoclasts	Whole-body bone mass ↑ (C57BL/6 mice)	[109]
	CaCO_3_ (rhombohedron-like/fusiform)	Osteogenesis	Dose- and shape-dependent osteogenesis effect Enhanced osteogenesis for promoting calcium ions	Alkaline phosphatase activity ↑ Collagen secretion ↑ Osteogenesis ↑Adipogenesis ↓ (hADSCs)	[110]
	CaCO_3_ (vaterite), methoxy poly(ethylene glycol)-block-poly(L-glutamic acid), doxorubicin,	Osteosarcoma	Increasing stability of doxorubicin loading by nanostructure pH-dependent release of drugs Biocompatibility	Anti-tumor effect Anti-bone destruction effect (BALB/c mice)	[111]
	CaCO_3_ (vaterite), alkaline phosphatase, vancomycin	Antibacterial effect	High surface area and porous structure for drug- and biomolecule-loading microcarriers Promotes bioactivity by releasing calcium ions	Cell attachment induction (MC3T3-E1) Antibacterial effect (S. *aureus*)	[112]
	CaCO_3_ (vaterite), ciprofloxacin	Osteomyelitis	Sustainable ciprofloxacin release through porous structure and biodegradability Uniformly entrapped ciprofloxacin by nanostructure	Antibacterial effect (methicillin-resistant and methicillin-susceptible S. *aureus*, *E. faecalis*, *A. baumannii*, *P. aeruginosa*, *E. coli*, and *K. pneumonia*; methicillin-resistant and methicillin-susceptible coagulase-negative *staphylococci*)	[113]
Coating	CaCO_3_ (calcite), CuS, polydopamine, Ti alloy screws	Antibacterial implant	Releasing calcium ions for bioactivity High surface area and porous structure for CuS loading by microsized template	Antibacterial effect (*S. aureus*, *E. coli*)	[114]
	CaCO_3_ (calcite, aragonite, vaterite), glutamate acid, dopamine, Mg alloy	Orthopedic implant	Increasing the surface roughness of template to enhance cell attachment	MC3T3-E1 proliferation, differentiation ↑ Osteogenesis promotion (SD rats)	[115]
	CaCO_3_ (calcite, vaterite), Ti6Al4V alloy	Antibacterial implant	Superhydrophilic template surface with reduced average surface roughness by nanostructure	Antibacterial effect by reducing bacterial adhesion on the template surface (*S. aureus*)	[116]
	CaCO_3_ (vaterite), Ti film	Bone implant	To improve biological response to osteoblast cells Increasing bone-like nodule formation on the surface	MC3T3-E1 proliferation and differentiation ↑ Osteogenesis promotion (MC3T3-E1)	[117]

Abbreviations: BMP-2, bone morphogenetic protein-2; hADSCs, human adipose-derived stem cells; PEG, polyethylene glycol; GelMA, methacrylated gelatin; BMSC, bone marrow mesenchymal stem cells; HA-MC, minocycline-modified hyaluronic acid; ZOL, zoledronic acid; PBAE-SA, (SA)-modified poly-β-amino ester; BMM, and bone marrow macrophages.

**Table 6 biomimetics-09-00511-t006:** Calcium-phosphate-based biomedical research for bone regeneration applications.

Biomedical Application	Materials	Primary Function	Ca-Based Effects	Biomedical Results	Ref.
Scaffold	Calcium phosphate (BCP), hydroxyethyl chitosan, polyvinyl alcohol	Bone regeneration	To promote hydrogel bonds, physical crosslinking, and biomineralization	Improved the compressive strength of the HECS/PVA/BCP hydrogel without sacrificing the porous structure Further improved cytocompatibility via the addition of HECS and in vitro biomineralization	[118]
	Calcium phosphate (BCP)	Bone tissue engineering	To promote osteoconductivity and bioresorbable	Presented comparable mechanical properties with human cancellous bone and higher cell proliferation rates (rat bone mesenchymal stem cells)	[119]
	Calcium phosphate (hydroxyapatite), chitosan, polyvinyl alcohol	Bone tissue engineering	To provide larger surface areas for ion exchange	Facilitated osteoblast cells to attach and proliferate (mouse osteoblast cells)	[120]
	Calcium phosphate (DCP), sodium alginate	Bone substitute	To provide an alternative option for PMMA	Increased cell (DPSCs) availability ratio, with no influence observed on cell shape, confirming the in vitro biocompatibility of the materials	[121]
	Calcium phosphate (hydroxyapatite), carbon nanotube	Bone substitute	To attract HPO_4_^2−^ by Ca^2+^ via electrovalent bonding to synthesize HA nanocrystals	Presented a Ca/P ratio of the apatite layer on the surface of the scaffold as 1.66, which was close to the ratio of normal bone	[122]
	Calcium phosphate (hydroxyapatite, TCP)	Bone regeneration	To adjust bio-performance by the HA/TCP ratio and pores	Presented an adjustable biodegradation rate by the HA/TCP ratio at an inverse relation, which is promising for designing patient-specific scaffolds	[123]
Drug delivery system	Calcium phosphate (hydroxyapatite), poly(ε-caprolactone)/poly(glycerol sebacate)	Bone tissue regeneration	To act as a simvastatin-loading nanocarrier	MC3T3E1 osteoblast cells/enhanced osteoblast cell growth, proliferation, and adhesion	[124]
	Calcium phosphate (hydroxyapatite), mesoporous silica material	Bone therapy	To act as a doxycycline-hydrochloride-loading nanocarriers	The 30%CaP@MSi allowing completion of 5-day release of the drug	[125]
	Calcium phosphate	Bone regeneration	Plasmid-DNA-encoding VEGF/siRNA inhibiting TNF-α-loading nanocarriers	Increased levels of bone-formation-related markers at the protein and gene levels in 3mixCaP after 10 days	[126]
Coating	Calcium phosphate (hydroxyapatite), titanium	Bone regeneration	To contribute to the control of cell adhesion and mineral binding	New bone beginning to develop at the implant interface after 2 weeks (rabbit femurs)	[127]
	Calcium phosphate, lipid nanoparticle	Bone therapy	To provide better cell accumulation than uncoated nanoparticles (NPs)	More dye delivered by CaP NPs to the cells within 24 h than the uncoated NPs	[128]
	Calcium phosphate (hydroxyapatite), graphene oxide	Bone tissue engineering	To find out its efficacy as an osteoinductive material	Catalyst for dye degradation and water treatment purposes	[129]
	Calcium phosphate (hydroxyapatite), molybdenum disulfide	Bone therapy and bone regeneration	To boost bone regeneration and integration around the implant	Exhibited adequate in vivo tissue compatibility and outstanding bone regeneration ability in the rat tibia defect model	[130]

**Table 7 biomimetics-09-00511-t007:** Calcium-silicate-based biomedical research for bone regeneration and healing.

Biomedical Application	Materials	Primary Function	Ca-Based Effects	Biomedical Results	Ref.
Scaffold	CaSi-ZrO_2_	Load-bearing implants	Improvement of mechanical biocompatibility Concentration-dependent antibiotic effect Promotion of osteogenic activity	Long-term stability Antibacterial ability against *E. coli* and *S. aureus* hMSC osteogenesis ↑	[133]
	Osteopontin motif-modified MCS	Bone regeneration scaffold	Bone-mimicking structure by mesoporosity Enhanced apatite formation by the large surface area	HUVEC adhesion and proliferation ↑ hBMSC osteogenic differentiation ↑ Vessel formation and bone growth ↑ in rabbits	[134]
	Lanthanum, MCS, chitosan	Bone defect repair	Proliferation and osteogenic differentiation by Ca^2+^ release	Cell adhesion, spreading, and proliferation ↑ of hBMSCs New bone formulation ↑ in rats	[135]
	Amorphous calcium silicate, titanium	Bone tissue engineering	Porosity suitable for bone tissue engineering Apatite deposition on the scaffold by the Ca_2+_ release and large surface area	Inhibition of rapid degradation Enhanced compressive strengths by MCS (no cell and in vivo data)	[136]
	α-Tricalcium phosphate, mesoporous calcium silicate nanoparticle	Bone regeneration cement	Decrease in inflammation by the alkaline dissolution reaction of MCS Mesoporosity-induced hydroxyapatite mineralization	Bone-like hydroxyapatite formation ability by mesoporosity New bone formulation ↑ in rats	[137]
Drug delivery system	Ginsenoside Rb1, polycaprolactone, MCS, calcium sulfate	Bone substitute scaffold	Uniform porous structure and suitable environment provided for cells	Anti-inflammation, depending on the drug concentration Cell proliferation and mineralization ↑ of hDPSCs Bone regeneration in rabbits	[138]
	Vancomycin, PLGA, Bredigite (Ca_7_MgSi_4_O_16_)	Bone tissue regeneration Local antibiotic Delivery	Drug loading and implementation template of porous scaffold	Local pH buffering by PLGA Sustained drug release Cytocompatibility ↑ of 3500 hDPSCs	[139]
	Triton-100, silver ion, MCS nanoparticle	Bone defect filling material	Sustained-release scaffold by interaction Biomineralization increase by release of Ca^2+^ and SiO_3_^2−^	Sustained release for 7 days Antibacterial effects against *E. faecalis* Low toxicity by MC3T3-E1	[140]
	BMP-2, MCS, calcium sulfate, polycaprolactone	Bone regeneration	Enhanced hydroxyapatite precipitation and crystallization Cell differentiation by released Ca and Si ions	Prolonged and controlled drug release over 6 months Proliferation and osteogenesis ↑ of hDPSCs Angiogenesis ↑ in rabbits	[141]
	FGF-2, MCS, polycaprolactone	Bone-healing composite filler	Apatite deposition on the MCS scaffold Enhancement of bone cell differentiation by sustained release of Ca and the drug	Cell proliferation and ALP activity ↑ of hWJMSCs Healing of femur bone defect in rabbist	[142]
	Chlorhexidine, MCS nanoparticle	Dental care biomaterial	Barrier layer formation for dentin by continuous apatite deposition Antibacterial activity by sustained release of chlorhexidine binding MCS’s Si ion	Sustained release Antibacterial activity against *E. faecalis* Low cytotoxicity by HDPCs Low dentin permeability and inflammation in rats	[143]
	Genistein, curcumin, Mg-CS, polyetheretherketone	Implant for bone substitutes	Increase surface roughness and wettability by mesoporous nanoparticles Stimulation of cell proliferation and differentiation by release of Ca, Mg, and Si ions	Apatite mineralization Cell adhesion and proliferation ↑ of rBMSCs Antibacterial activity against *E. coli* and *S. aureus* Osteogenesis and osseointegration in rabbits	[144]
Coating	Zinc-modified CS, polycaprolactone, graphene oxide	Orthopedic implant	Bone-like apatite growing on the implant surface by forming amorphous Ca	Antibacterial activity Cell viability and differentiation ↑ of MG63 human osteoblast cells	[145]
	Europium, calcium silicate, titanium	Biomedical implant coating	Enhanced wettability by hydrophilicity of CS Improvement of apatite formation on the titanium implant by Ca release	Biologically similar apatite-forming ability Cell adhesion, proliferation, and ALP activity ↑ of hFOB	[146]

Abbreviations: ALP, alkaline phosphatase; BMP-2, bone morphogenetic protein-2; FGF-2, basic fibroblast growth factor; *E. coli*, *Escherichia coli*; *E. faecalis*, *Enterococcus faecalis*; hBMSCs, human bone marrow stromal cells; hDPSCs, human dental pulp stem cells; hDPCs, human dental pulp cells; hFOB, human fetal osteoblastic cell lines; hMSCs, human mesenchymal stem cells; HUVECs, human umbilical vein endothelial cells; hWJMSCs, human Wharton’s jelly mesenchymal stem cells; MCS, mesoporous calcium silicate; PLGA, polylactic-co-glycolic acid; rBMSCs, rat bone mesenchymal stem cells; *S. aureus*, *Staphylococcus aureus*.

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
