# Peer review of "Biomimetic Scaffolds of Calcium-Based Materials for Bone Regeneration"

_biomimetics, 2024, doi:10.3390/biomimetics9090511_

Round 1

Reviewer 1 Report

Comments and Suggestions for Authors

The paper's title "Biomimetic Scaffolds of Calcium-based Nanoporous Materials for Bone regeneration" suggested potential as a timely and valuable review paper. A scan of the literature sourced suggested modern work had been cited and this was also encouraging. Unfortunately however I had multiple concerns with the contents of the manuscript, there were numerous errors, numerous emissions and I don’t feel any focus on ‘nanoporous’ (tunable nanostructures are mentioned in the abstract, but then aren’t in the bulk of the manuscript and nanostructures aren’t the same as nanoporous). The manuscript does not show sufficient understanding of either bone repair and regeneration or the calcium-based materials that are utilised and consequently I simply do not feel it is suitable for publication.

This lack of understanding can be highlighted through specific points raised through the script:

For example:

In the introduction it is stated that biomaterial-based strategies have emerged as viable alternatives to traditional approaches – what are traditional approaches – biomaterials in various forms have been applied for centuries.

Line 30-31 The combination of calcium and phosphorus  isn’t a composite

Line 56 – this statement is incredibly simplistic and fundamentally flawed – phosphate is not an element and you simply cannot state that ‘calcium and phosphates elements induce the formation of new bones’

Line 75 – “Calcium-based nanoporous materials, such as calcium carbonate, calcium phosphate, 75 and calcium silicate, encompass a variety of nanostructured materials primarily composed of calcium. These materials exhibit nanoscale porosity, imparting unique physical,  chemical, or mechanical properties.” – nanostructured is not purely nanoporous and there is no discussion throughout the manuscript of how nanoporous structures are achieved.

In table 1 it appears to be suggested that Calcium silicate and carbonate are neither biocompatible nor biodegradable and why is only Tri calcium phosphate considered as a calcium phosphate in this table

When it comes to the discussion of calcium carbonate the focus is solely on precipitation and is incorrectly referenced -  reference 43 does not appear to mention a ‘biomedical synthesis method nor organic compounds as templates’ and more broadly there is no mention of calcium carbonate as an abundant mineral or it’s derivation from waste material. Table 2 should be referenced, description of synthesis methods are simplistic and where is the focus on nanoporous.

The section on calcium phosphates focuses on the precipitation of nanoparticles and not nanoporous material – how is this material ultilised in the formation of nanoporous structures?

In discussion of calcium silicates, applications in thermal insulation have nothing to do with bone regeneration.

When it comes to the discussion around applications of the various calcium-based materials, there is again not a focus on nanoporous materials and some flawed science.

Additionally when it comes to the discussion of Calcium silicates – line 397, you simply cannot say that ‘In contrast to calcium carbonate and calcium phosphate, calcium silicate has been studied in a variety of in vivo models that have confirmed its effectiveness – there is masses of in vivo data on calcium phosphates particularly.

Fundamentally the paper title and abstract do not adequately represent the contents of the paper, the content is confusing and does not ultimately add significantly to the mass of literature pre-existing on calcium-based materials for bone repair and regeneration.

Comments on the Quality of English Language

Use of English language is largely acceptable throughout the paper although perhaps some of the concerns around the content may be due to challenges with the language.

Author Response

The paper's title "Biomimetic Scaffolds of Calcium-based Nanoporous Materials for Bone regeneration" suggested potential as a timely and valuable review paper. A scan of the literature sourced suggested modern work had been cited and this was also encouraging. Unfortunately, however I had multiple concerns with the contents of the manuscript, there were numerous errors, numerous emissions and I don’t feel any focus on ‘nanoporous’ (tunable nanostructures are mentioned in the abstract, but then aren’t in the bulk of the manuscript and nanostructures aren’t the same as nanoporous). The manuscript does not show sufficient understanding of either bone repair and regeneration or the calcium-based materials that are utilized and consequently I simply do not feel it is suitable for publication. This lack of understanding can be highlighted through specific points raised through the script:

- Answer: Thank you for your insightful comments. We understand the reviewer's concerns and totally agree that emphasizing the nanoporosity without sufficient content is inappropriate.  In the revision, we removed such inappropriate expressions (including title, sentences and so on) accordingly, considering the reviewer's valuable comments, and thoroughly improved the lacking discussion sections. We apologize for the shortcomings of the first version, in this regard. We did our best for the revision. Our paper aimed to provide a comprehensive review of the latest trends in varous calcium-based biomaterials used for bone tissue regeneration. So, our manuscript covers three representative Ca-based materials, (1) calcium carbonate, (2) calcium phosphate, and (3) calcium silicate, along with their latest technological advancements. Dealing with all three types of Ca-based mateirals at the same time could be distinctive points from the other reviews. We kindly ask the reviewer to consider this aspect in your evaluation.

For example:

  1. In the introduction it is stated that biomaterial-based strategies have emerged as viable alternatives to traditional approaches – what are traditional approaches – biomaterials in various forms have been applied for centuries.

- Answer: Thanks for your comments. We agree with your comment. We edited sentence (line 27-30).

  1. Line 30-31 The combination of calcium and phosphorus isn’t a composite

- Answer: We agree with your comment. We made mistakes. We edited sentence (line 30-33).

  1. Line 56 – this statement is incredibly simplistic and fundamentally flawed – phosphate is not an element and you simply cannot state that ‘calcium and phosphates elements induce the formation of new bones

- Answer: We agree with your comment. We revised the sentence for clarity (line 57-58).

Strictly speaking, calcium and phosphate elements themselves do not directly 'induce' the formation of new bones. Instead, they play an important role in the bone formation process. These elements are essential minerals used as building blocks for new bone tissue. Therefore, the term 'induce' might be somewhat exaggerated.

To be more accurate, it would be better to say that 'calcium and phosphate elements assist in the formation of new bones’.

  1. Line 75 – “Calcium-based nanoporous materials, such as calcium carbonate, calcium phosphate, and calcium silicate, encompass a variety of nanostructured materials primarily composed of calcium. These materials exhibit nanoscale porosity, imparting unique physical,  chemical, or mechanical properties.”

– nanostructured is not purely nanoporous and there is no discussion throughout the manuscript of how nanoporous structures are achieved.

- Answer: We agree with your comment. We made mistakes. We have revised all words of nanoporous in the manuscript.

  1. In table 1 it appears to be suggested that Calcium silicate and carbonate are neither biocompatible nor biodegradable and why is only Tri calcium phosphate considered as a calcium phosphate in this table.

- Answer: Thanks for your comments. We edited the table. Table 1 does not suggest that calcium silicate and calcium carbonate are neither biocompatible nor biodegradable. It means that they are the same as calcium phosphate.

Also, Calcium phosphate has several compounds or type, but Ca3(PO4)2 was selected as the representative one.

  1. When it comes to the discussion of calcium carbonate the focus is solely on precipitation and is incorrectly referenced -  reference 43 does not appear to mention a ‘biomedical synthesis method nor organic compounds as templates’ and more broadly there is no mention of calcium carbonate as an abundant mineral or it’s derivation from waste material. Table 2 should be referenced, description of synthesis methods are simplistic and where is the focus on nanoporous.

- Answer: Thanks for your comments. We agree with your comment. We corrected incorrect references, added new ones, detailed the description of the synthetic method, and removed the focus on nanoporous materials.

  1. The section on calcium phosphates focuses on the precipitation of nanoparticles and not nanoporous material – how is this material ultilised in the formation of nanoporous structures?

 - Answer: Thanks for your comments. Following the previous comments, the expression nanoporous has been removed and replaced with the expression nanostructure.

  1. In discussion of calcium silicates, applications in thermal insulation have nothing to do with bone regeneration.

 - Answer: Thanks for your comments. We deleted the content.

  1. When it comes to the discussion around applications of the various calcium-based materials, there is again not a focus on nanoporous materials and some flawed science.

- Answer: Thanks for your comments. We agree with your comment. We have aligned the title and manuscript to avoid confusion. Also, We have revised content by referring to the comments mentioned.

  1. Additionally, when it comes to the discussion of Calcium silicates – line 397, you simply cannot say that ‘In contrast to calcium carbonate and calcium phosphate, calcium silicate has been studied in a variety of in vivo models that have confirmed its effectiveness – there is masses of in vivo data on calcium phosphates particularly.

- Answer: We agree with your comment. We agree with your comment. We made mistakes. We edited sentence.

  1. Fundamentally the paper title and abstract do not adequately represent the contents of the paper, the content is confusing and does not ultimately add significantly to the mass of literature pre-existing on calcium-based materials for bone repair and regeneration.

- Answer: Thanks for your comments. We agree with your comment. We have aligned the title and manuscript to avoid confusion. Also, We have revised and supplemented the content by referring to the comments mentioned.

Reviewer 2 Report

Comments and Suggestions for Authors

The manuscript reviews the “Biomimetic Scaffolds of Calcium-based Nanoporous Materials for Bone regeneration”. It is well-written and can be accepted for publication after minor revision. Some suggestions are given below.

1.  Though these biomimetic scaffolds typical exhibit nano-sized porous structures, “nanoporous” in the title may still be misleading.

2.       There are 27 pages in the manuscript, a table of content may be useful.

3.       The last three figures concerning the biomedical applications of calcium-based materials are focused on scaffold, drug delivery, and coatings. Are these figures from the references? Some explanation should be given in the caption to avoid confusion.

Author Response

The manuscript reviews the “Biomimetic Scaffolds of Calcium-based Nanoporous Materials for Bone regeneration”. It is well-written and can be accepted for publication after minor revision. Some suggestions are given below.

  1. Though these biomimetic scaffolds typical exhibit nano-sized porous structures, “nanoporous” in the title may still be misleading.

- Answer: Thank you for your comment. We edited title.

  1. There are 27 pages in the manuscript, a table of content may be useful.

- Answer: Thank you for your comment. It may be useful to include a table of content, but very few review papers include a table of contents. So, we decided to include an overview of the structure of the manuscript. We sincerely appreciate your suggestion once again.

  1. The last three figures concerning the biomedical applications of calcium-based materials are focused on scaffold, drug delivery, and coatings. Are these figures from the references? Some explanation should be given in the caption to avoid confusion.

- Answer: Thank you for your comment. We have thoroughly referenced each research paper, but all the figures included in the manuscript have been created by the authors themselves, without using any figures from the references. Also, We added an explanation to each caption.

Reviewer 3 Report

Comments and Suggestions for Authors

This review article explores the potential of calcium-based materials such as calcium carbonate, calcium phosphate, and calcium silicate in bone tissue engineering. These materials exhibit high biocompatibility, tunable nanostructures, and biodegradability, making them ideal for regenerative medicine applications. The article details the synthesis methods, properties, and biomedical applications of these materials, highlighting their role in enhancing cell adhesion, proliferation, differentiation, and controlled drug release. The review also discusses the future directions and multidisciplinary applications of these nanoporous materials in bone regeneration. However, kindly address minor comments prior to publication.

1. How do the mechanical properties of these scaffolds compare to those of natural bone?

2. What are the limitations of the current synthesis methods for these materials?

3. Are there any known long-term biocompatibility issues with these materials?

4. How does the porosity of these materials influence their effectiveness in bone regeneration?

5. How do these materials interact with different types of cells involved in bone regeneration?

6. How do the different crystalline forms of calcium carbonate affect their performance in bone regeneration?

Author Response

This review article explores the potential of calcium-based materials such as calcium carbonate, calcium phosphate, and calcium silicate in bone tissue engineering. These materials exhibit high biocompatibility, tunable nanostructures, and biodegradability, making them ideal for regenerative medicine applications. The article details the synthesis methods, properties, and biomedical applications of these materials, highlighting their role in enhancing cell adhesion, proliferation, differentiation, and controlled drug release. The review also discusses the future directions and multidisciplinary applications of these nanoporous materials in bone regeneration. However, kindly address minor comments prior to publication.

  1. How do the mechanical properties of these scaffolds compare to those of natural bone?

- Answer: Thanks for your comments. The content mentioned in the comments is detailed in Sections 2 in the manuscript.

  1. What are the limitations of the current synthesis methods for these materials?

- Answer: Thanks for your comments. The content mentioned in the comments is detailed in Sections 2 in the manuscript.

  1. Are there any known long-term biocompatibility issues with these materials?

- Answer: Thanks for your comments. The content mentioned in the comments is detailed in Sections 4 in the manuscript.

  1. How does the porosity of these materials influence their effectiveness in bone regeneration?

- Answer: Thanks for your comments. The content mentioned in the comments is detailed in Sections 4 in the manuscript.

  1. How do these materials interact with different types of cells involved in bone regeneration?

- Answer: Thanks for your comments. The content mentioned in the comments is detailed in Sections 3 in the manuscript.

  1. How do the different crystalline forms of calcium carbonate affect their performance in bone regeneration?

- Answer: Thanks for your comments. The content mentioned in the comments is detailed in Sections 3.1 in the manuscript.

Round 2

Reviewer 1 Report

Comments and Suggestions for Authors

The authors have really taken on board the serious concerns raised with the first submission and I was very pleasantly surprised with this resubmission. The title and abstract are now relevant to the bulk of the text and all major concerns have been addressed. I am happy with some minor edits to consider the work suitable for publication.

Line 51-53 – I would remove the specific focus on nanoparticles – as this is no longer relevant to the title – sufficient to say calcium carbonate facilitate receptor interactions and promote cell differentiation and proliferation.

Line 77 – Calcium based materials for biomaterial application include calcium carbonates, silicates and phosphates. These materials can exhibit nanoscale structures or nanoporosity which are able to impart unique physical, chemical or mechanical properties and specific functionalities. Owing to tailored biodegradability and excellent biocompatibility, calcium-based materials have been extensively studied for various biomaterial scaffolds applications.

Line 152 – Remove “Calcium phosphate materials have similar advantages to calcium phosphate-based materials”

Line 156 – add scaffolds after calcium phosphate (calcium phosphate doesn’t by itself have high surface area)

Section 4 – Valuable content but insufficiently referenced – reference at the end of each sentence even if several sentences rely on the same reference, either this or quote directly.

Comments on the Quality of English Language

I would recommend a last read through by someone with native English proficiency.

Author Response

# Minor Revisions

The authors have really taken on board the serious concerns raised with the first submission and I was very pleasantly surprised with this resubmission. The title and abstract are now relevant to the bulk of the text and all major concerns have been addressed. I am happy with some minor edits to consider the work suitable for publication.

- Answer: Thank you for your insightful comments.

  1. Line 51-53 – I would remove the specific focus on nanoparticles – as this is no longer relevant to the title – sufficient to say calcium carbonate facilitate receptor interactions and promote cell differentiation and proliferation.

- Answer: Thanks for your comments. We agree with your comment. We deleted sentence (line 51-53).

  1. Line 77 – Calcium based materials for biomaterial application include calcium carbonates, silicates and phosphates. These materials can exhibit nanoscale structures or nanoporosity which are able to impart unique physical, chemical or mechanical properties and specific functionalities. Owing to tailored biodegradability and excellent biocompatibility, calcium-based materials have been extensively studied for various biomaterial scaffolds applications.

- Answer: Thanks for your comments. We modified the paragraph as your recommendation. (line 75-80).

  1. Line 152 – Remove “Calcium phosphate materials have similar advantages to calcium phosphate-based materials”

- Answer: Thanks for your comments. We deleted sentence.

  1. Line 156 – add scaffolds after calcium phosphate (calcium phosphate doesn’t by itself have high surface area)

- Answer: Thanks for your comments. We added scaffolds after calcium phosphate.

  1. Section 4 – Valuable content but insufficiently referenced – reference at the end of each sentence even if several sentences rely on the same reference, either this or quote directly.

- Answer: Thanks for your comments. We agree with your comment. We added several references.